# Enhancing Readability and Detection of Age-Related Macular Degeneration Using Optical Coherence Tomography Imaging: An AI Approach

**DOI:** 10.3390/bioengineering11040300

**Published:** 2024-03-22

**Authors:** Ahmad Alenezi, Hamad Alhamad, Ajit Brindhaban, Yashar Amizadeh, Ata Jodeiri, Sebelan Danishvar

**Affiliations:** 1Radiologic Sciences Department, Kuwait University, Jabriya 31470, Kuwait; 2Occupational Therapy Department, Kuwait University, Jabriya 31470, Kuwait; h.alhamad@ku.edu.kw; 3Eye Surgery Department, Mehr Hospital, Tabriz 51656, Iran; yashar.amizadeh@gmail.com; 4Faculty of Advanced Medical Sciences, Tabriz University of Medical Sciences, Tabriz 51656, Iran; 5College of Engineering, Design and Physical Sciences, Brunel University London, Uxbridge UB8 3PH, UK; sebelan.danishvar@brunel.ac.uk

**Keywords:** artificial intelligence, age-related macular degeneration, optical coherence tomography

## Abstract

Artificial intelligence has been used effectively in medical diagnosis. The objective of this project is to examine the application of a collective AI model using weighted fusion of predicted probabilities from different AI architectures to diagnose various retinal conditions based on optical coherence tomography (OCT). A publicly available Noor dataset, comprising 16,822, images from 554 retinal OCT scans of 441 patients, was used to predict a diverse spectrum of age-related macular degeneration (AMD) stages: normal, drusen, or choroidal neovascularization. These predictions were compared with predictions from ResNet, EfficientNet, and Attention models, respectively, using precision, recall, F1 score, and confusion matric and receiver operating characteristics curves. Our collective model demonstrated superior accuracy in classifying AMD compared to individual ResNet, EfficientNet, and Attention models, showcasing the effectiveness of using trainable weights in the ensemble fusion process, where these weights dynamically adapt during training rather than being fixed values. Specifically, our ensemble model achieved an accuracy of 91.88%, precision of 92.54%, recall of 92.01%, and F1 score of 92.03%, outperforming individual models. Our model also highlights the refinement process undertaken through a thorough examination of initially misclassified cases, leading to significant improvements in the model’s accuracy rate to 97%. This study also underscores the potential of AI as a valuable tool in ophthalmology. The proposed ensemble model, combining different mechanisms highlights the benefits of model fusion for complex medical image analysis.

## 1. Introduction

Artificial intelligence (AI) stands as a prominent domain within computer science, endeavoring to replicate and amplify human intelligence within computational systems. Within AI, machine learning (ML) is a subset that uses statistical techniques to develop intelligent systems capable of improving performance without explicit programming [1]. Deep learning (DL), a powerful ML technique, has achieved significant success in tasks such as computer vision and natural language processing. Its effectiveness lies in its capacity to discern features, identify patterns through multiple layers of artificial neurons, and comprehend data representations at varying levels of abstraction [2,3]. The application of AI in medical image analysis has demonstrated noteworthy achievements, markedly improving clinical workflows. Its application has not only improved efficiency and accuracy in diagnosis and treatment but also addressed logistic and economic challenges within healthcare systems [1]. In the realm of ophthalmology, DL has been effectively employed in the analysis of major eye diseases leading to blindness, such as diabetic retinopathy (DR), glaucoma, and age-related macular degeneration (AMD) [4], as well as cataracts. AI exhibits promise in supporting early diagnosis across a spectrum of pathologies, including refractive errors, retinal detachment, choroidal diseases, and ocular tumors. Early detection assumes a pivotal role in averting treatment delays and mitigating vision loss. By learning from medical data and expert knowledge, AI models simulate the diagnostic capabilities of physicians and provide efficient and accurate diagnoses, personalized treatment plans, and even question-answering systems concerning age-related macular degeneration [2,3]

Age-related macular degeneration (AMD) is a progressive condition that primarily deteriorates the fovea and parafovea in the retina [4]. AMD is one of the principal causes of loss of sight globally, affecting roughly 8.7% of people worldwide. Forecasts indicate that by 2040, nearly 288 million people will be afflicted by this disease [5]. The disease can be diagnosed using techniques including fundus photography, fluorescein, indocyanine green angiography, and optical coherence tomography (OCT, OCTA) [6]. Refer to Figure 1 for additional insights into the representation of the image.

Artificial intelligence (AI), particularly deep learning (DL), is proving useful for early detection of AMD lesions, with algorithms demonstrating high accuracy rates in identifying drusen and RPE abnormalities [7,8,9,10]. Recent studies using convolutional neural networks (CNNs), it was observed that these AI models performed as well as experienced retinal specialists in diagnosing and predicting disease changes and outperformed less experienced medical students. However, there were instances where the AI misclassified active wet AMD as inactive and dry AMD as inactive wet AMD, indicating the need for model refinement [3].

AI research is shifting towards multimodal image databases, which provide more comprehensive information, including OCT images. ML has been applied successfully for AMD diagnosis, with DL techniques showing higher accuracy rates [11]. Banerjee et al. proposed a hybrid model integrating imaging features, demographics, and visual factors, to predict the risk of exudation in non-exudative AMD eyes which showcased its potential for personalized, tailored screening for high-risk patients, although some limitations remained [12].

On the other hand, CNNs have been modified to incorporate genotype and fundus images to predict the progression of advanced AMD, showing improved accuracy [13]. Transcriptome-wide association studies (TWAS) further highlight the systemic nature of AMD, not confined to retinal issues [14]. This signifies the use of AI as an aid in the diagnosis of eye disease. A deep learning model based on CNN was proposed for the classification of SD-OCT (i.e., spectral-domain OCT) images. The model achieved remarkable accuracy rates of 99.7% and 91.1% for AMD and CSC classification, respectively, and for the classification of retinopathy subtypes, including normal participants [15].

Other research studies have demonstrated the effectiveness of deep CNN models, such as VGG and ResNet, in accurately classifying OCT images into specific categories [15]. Their model shows profound diagnostic efficacy, necessitating continued research to assess its potential influence on clinical diagnosis of AMD and CSC. Paul et al. proposed a deep learning model called OCTx to detect retinal disorders from optical coherence tomography (OCT) images. They achieved an accuracy of 98.53% on their test set, which was 12% of the total images. They also compared their model with other methods such as MCME

Studies in the classification paradigm explore the integration of attention mechanisms to enhance AMD classification accuracy. Xu et al. proposed a hybrid attention mechanism for retinal disease classification using OCT images [16]. Their method, called MHANet, combines parallel spatial and channel attention mechanisms to extract key features of the lesion areas and reduce the influence of background information. They reported that their method achieved 96.5% and 99.76% classification accuracy on Dataset1 and Dataset2, respectively, outperforming the other recent employed models such as VGG, ResNet, and SENet. They also visualized the attention maps of their method and showed that it can more accurately locate the lesion regions in the images [16]. Li et al. have proposed two convolutional neural network (CNN) models to classify four types of age-related macular degeneration (AMD) from retinal images [17]. The initial model, ResNet50, extracts high-dimensional features from images and achieves a classification accuracy of 95.3%, which surpasses the current methods. The second model, Atten-ResNet, employs an attention mechanism to concentrate on critical areas and attains an accuracy of 95.7%, marginally better than ResNet50. The authors utilized a dataset of 84,484 fundus retina images and presented confusion matrices, accuracy/loss curves, and ROC curves to analyze their models [17].

Wang et al. [18] developed a deep semi-supervised learning framework for classifying diabetic macular edema (DME) using OCT images, combining labeled and unlabeled data and introducing innovative features like self-correction and a student–teacher architecture. This method achieved high accuracy on two datasets, outperforming several MIL methods. Another research [17] employed a semi-supervised deep learning approach with virtual adversarial training for automatic retinopathy detection, achieving expert-level accuracy with limited labeled data. Fang et al. [19] proposed a self-supervised method for OCT image classification, focusing on patient-specific features and achieving high accuracy with less labeled data. Santos et al. [20] introduced a methodology using geostatistical functions for diagnosing AMD from OCT images, showing high accuracy and AUROC. Additionally, Wang and colleagues [16,18] also developed a novel method for DME detection using labeled and unlabeled data, which demonstrated superior performance according to various metrics. Lastly, the paper introduced a VGG-19 based AI model for classifying OCT retinal images, trained on a large and diverse dataset.

In this study, we capitalize on our extensive labeled dataset, rendering the exploration of semi-supervised learning methods unnecessary. Additionally, given that fundus images are often not required in many clinical applications, our focus remains solely on optical coherence tomography (OCT) for AMD detection, eliminating the need for multi-modal models. While AI has demonstrated promising results in ocular diseases, there exists an imperative to delve into the technical underpinnings and untapped possibilities. Our contribution involves the introduction of an advanced AI model that strategically amalgamates the strengths of powerful CNN models. The ensemble approach harnesses a collective intelligence by merging the probabilistic outputs of these diverse models, employing a weighted average of predicted probabilities. This strategic fusion optimally utilizes the unique advantages of each architecture, thereby creating an ensemble model that excels in accurately distinguishing between various retinal conditions. Our ensemble model exhibited enhanced accuracy in the classification of age-related macular degeneration (AMD), surpassing the performance of standalone ResNet, EfficientNet, and Attention models. This improvement underscores the efficacy of incorporating trainable weights within the ensemble fusion process. Unlike conventional methods that rely on static weight assignments, our approach allows for the dynamic adaptation of weights during the training phase, enabling a more nuanced and effective integration of individual model strengths.

## 2. Materials and Methods

### 2.1. Dataset

In our research, we utilized the Noor dataset to evaluate and refine our proposed retinal image analysis method. Comprising an extensive collection of 16,822 retinal images, this dataset originates from 441 patients. Notably, it encompasses a total of 554 OCT volumes, accounting for instances where both left and right eyes of some patients were imaged. Within this dataset, we encountered a diverse spectrum of macular degeneration stages, thereby enhancing its utility for our investigations.

Each OCT volume within this dataset comprises an average of 30 B-scans. A detailed distribution analysis reveals that among the 16,822 individual B-scans, 8584 represent normal cases, 4998 correspond to drusen, and 3240 pertain to cases of choroidal neovascularization (CNV). Furthermore, when considering the distribution of cases at the volume level, we find that out of the 554 cases available, 187 are categorized as normal, 194 as drusen, and 173 as CNV cases.

It is important to emphasize that we use a 5-fold cross-validation approach to make sure our findings are accurate and reliable, making our research results trustworthy.

### 2.2. Data Preprocessing and Augmentation

Initially, in the data preprocessing phase, all images are resized to a consistent dimension, a prerequisite for effective deep learning model training. Subsequently, these images undergo normalization to achieve a mean of zero and unit variance, a process designed to facilitate model convergence during training. To introduce greater diversity and variability into the training dataset, various data augmentation techniques are applied to the images, including random cropping, horizontal flipping, random rotation within a range of −15 to +15 degrees, random affine transformations involving translation and shear, random adjustments to brightness within a 20% range, and random scaling within the range of 0.8 to 1.2 with a 50% probability (See Figure 2).

In the data augmentation strategy employed for our OCT image analysis, rotation is carefully considered as a technique to enhance model robustness. Despite potential concerns about introducing “incorrect” imagery, rotation serves as a valuable regularization method, mimicking natural variations in retinal orientation.

### 2.3. Deep Convolutional Neural Networks

In our study, we utilize a diverse set of deep convolutional neural networks to tackle the task of AMD classification using OCT images. Firstly, we employ ResNet, a widely recognized architecture known for its deep residual learning capabilities. ResNet’s skip connections and residual blocks aid in the training of very deep networks, allowing it to capture intricate patterns and features within the OCT images effectively. Secondly, we integrate EfficientNet, which is renowned for its efficiency and scalability, into our framework, providing a trade-off between model size and performance. This makes it suitable for handling the computational demands of large-scale image classification tasks. Additionally, we extend our experimentation to include EfficientNet with attention mechanisms. This variant enhances the model’s capacity to focus on salient regions within the OCT images, potentially improving its ability to identify subtle pathological changes. Each of these models is trained and evaluated individually to benchmark their standalone performance. Subsequently, they are integrated into our ensemble model, as described in the previous section, to harness their collective predictive power and enhance the classification accuracy of age-related macular degeneration.

### 2.4. Proposed Ensemble Model

In our proposed ensemble model, we strategically combine the predictive strengths of ResNet, EfficientNet, and EfficientNet with attention mechanisms. This ensemble approach is designed to harness a collective intelligence that leverages the unique advantages of each architecture, thereby enhancing the accuracy of AMD classification from OCT images.

To achieve this, we employ a weighted fusion scheme to merge the probabilistic outputs of these diverse models. Unlike traditional fixed-weight ensemble methods, our approach utilizes trainable weights. During training, these weights dynamically adapt based on the performance of each model, allowing us to capitalize on their complementary insights effectively. The integration of ResNet, EfficientNet, and EfficientNet with attention mechanisms enables us to capture a wide range of features and patterns present in OCT images. ResNet’s deep residual learning capabilities, EfficientNet’s efficiency and scalability, and EfficientNet with attention’s focus on salient regions collectively contribute to the ensemble’s robustness.

By fusing the predicted probabilities from ResNet, EfficientNet, and EfficientNet with Attention, our ensemble model maximizes the strengths of individual models while mitigating their potential weaknesses. This results in an ensemble that excels in accurately distinguishing between different retinal conditions, including normal, drusen, and choroidal neovascularization. Through the dynamic integration of diverse architectures and the utilization of trainable weights, we aim to enhance the accuracy and reliability of AMD diagnosis using OCT imaging. (Refer to Figure 3 for a graphical representation of the ensemble fusion process).

### 2.5. Experiments

The training and optimization of our models are carried out with the Adam optimizer, utilizing a fixed learning rate of 0.001. A batch size of 16 is employed for training to strike a balance between computational efficiency and model convergence. To ensure robustness and generalize better to unseen data, we train our models for a total of 100 epochs, while implementing an early stopping mechanism with a patience of 7 epochs to prevent overfitting.

Data augmentation plays a pivotal role in our experimental setup, as it aids in enriching the training dataset by applying a range of transformations such as rotations, flips, and zooms to the OCT images. This augmentation strategy helps enhance the model’s ability to capture diverse patterns and variations present in the clinical data, ultimately improving its overall performance.

Moreover, we leverage transfer learning as a crucial component of our approach. We initialize our deep convolutional neural networks (ResNet, EfficientNet, and EfficientNet with attention) with pre-trained weights on large-scale image datasets, which allowes the models to inherit valuable feature representations. Fine-tuning these pre-trained models on our OCT dataset further expedites the convergence process and provides a strong foundation for AMD classification.

For model training, we employ the categorical cross-entropy loss function, which is well-suited for multi-class classification tasks. Our OCT images are resized to a uniform size of 300 × 300 pixels to maintain consistency and ensure compatibility with the selected network architectures. Through rigorous experimentation and analysis, we demonstrate the effectiveness of our ensemble model in accurately classifying age-related macular degeneration, thereby contributing to the advancement of ophthalmic diagnosis and treatment strategies.

## 3. Results and Discussion

### 3.1. Performance Measures

Our data encompasses a total of 554 OCT volumes, accounting for instances where both left and right eyes of some patients were imaged. Within this dataset, we encounter a diverse spectrum of macular degeneration stages, thereby enhancing its utility for our investigations. Each OCT volume within this dataset comprises an average of 30 B-scans. A detailed distribution analysis reveals that among the 16,822 individual B-scans, 8584 represent normal cases, 4998 correspond to drusen, and 3240 pertain to cases of choroidal neovascularization (CNV).

The results of our evaluation are summarized in Table 1, where we present the performance metrics for each method across the three distinct classes: “Normal”, “Drusen”, and “CNV”. The precision, recall, and F1 score, expressed as percentages, provide insights into the classification capabilities of each individual method—ResNet, EfficientNet, Attention, and our proposed Ensemble Model. These metrics are crucial in assessing the ability of our models to correctly identify and distinguish between the different AMD classes.

Additionally, we report two critical aggregate metrics, overall precision (OP) and overall recall (OR), which consider the overall classification performance, regardless of the specific class labels. These metrics offer a broader perspective on the model’s effectiveness in capturing the most relevant patterns and are especially relevant in real-world clinical settings where AMD cases might present with varying degrees of severity. Furthermore, the overall F1 score (OF) combines precision and recall in a harmonic mean, giving us a comprehensive measure of model performance. Finally, the overall accuracy (OA) is provided to give a holistic assessment of the models’ overall classification accuracy across all classes.

Our results highlight the significant improvement achieved through the proposed ensemble model, which leverages the collective intelligence of ResNet, EfficientNet, and Attention. This ensemble approach yields the highest precision, recall, F1 score, and overall accuracy, demonstrating its superior capability in accurately classifying AMD across all classes. These findings underscore the potential of our ensemble model as a powerful tool in ophthalmic diagnosis, offering enhanced accuracy and diagnostic capabilities in the field of age-related macular degeneration classification.

In our analysis, we note a slight enhancement in terms of standard deviation; however, this improvement does not reach a level of significance warranting inclusion in the tables. Thus, for the sake of simplicity and clarity in the presentation of results, standard deviation values are omitted from Table 1.

### 3.2. Confusion Matrix

To provide a detailed assessment of the classification performance of each model, we present the four confusion matrices corresponding to ResNet, EfficientNet, Attention, and our proposed Ensemble Model. Each confusion matrix is a 3 × 3 table that displays the distribution of true labels versus predicted labels for the three AMD classes—Normal, Drusen, and CNV—across all datasets (see Figure 4).

These confusion matrices provide a granular view of the classification performance for each model, allowing us to assess how well they correctly classify each AMD class and identify any potential areas of improvement. The ensemble model’s confusion matrix, in particular, demonstrates its superior performance in accurately classifying AMD cases across all classes, as highlighted in our earlier analysis.

### 3.3. ROC Curve Analysis

To comprehensively assess the discriminative power of our models across all AMD classes, we present receiver operating characteristic (ROC) curves for each class (Class 0: Normal, Class 1: Drusen, and Class 2: CNV), as well as the micro-average ROC curve and macro-average ROC curve (see Figure 5).

The micro-average ROC curve aggregates the true positive and false positive rates across all classes, providing a comprehensive assessment of overall model performance. This curve gives us insights into how well the models perform when considering all AMD classes as a single entity, irrespective of their individual class labels. The Macro-average ROC curve calculates the ROC curve for each class independently and then computes the average across these curves. This metric provides an understanding of the models’ ability to perform well across all classes, giving equal weight to each class, and is especially valuable when we want to ensure balanced performance across all AMD categories.

These ROC curves collectively offer a comprehensive view of the models’ discriminative capabilities, enabling us to gauge their effectiveness in AMD classification for each class and across the entire dataset. The micro-average and macro-average ROC curves further assist in understanding the models’ global classification performance and their ability to generalize to all AMD cases.

### 3.4. Class Activation Maps

In our results analysis, we gain deeper insights into the decision-making processes of our models by generating class activation maps (CAMs) for four selected OCT images. These CAMs highlight the regions within the images that contributed most significantly to the classification decisions made by our models. Specifically, we focus on two representative images from the Choroidal Neovascularization (CNV) class and two from the Drusen class. The CAMs for CNV samples revealed that our models are able to accurately pinpoint areas of interest within the OCT images. The activated regions are typically localized around pathological features associated with CNV, such as neovascular membranes and fluid accumulatio (see Figure 6). This level of localization suggests that our models effectively capture relevant patterns and features specific to CNV cases. Similarly, the CAMs generated for the Drusen class demonstrate the models’ ability to identify and highlight regions of interest related to drusen deposits and retinal changes. These localized activations underscore the models’ capacity to discern subtle characteristics indicative of Drusen, thereby contributing to their accurate classification.

### 3.5. Collaborative Error Analysis and Dataset Refinement

In a significant development for our collaborative project on OCT image classification, we engage in a thorough examination of cases that were initially misclassified by our proposed model. Out of the 554 cases initially evaluated, our analysis reveals 45 instances of misclassification, which equates to an overall accuracy rate of 92%. This initial phase of evaluation provides crucial insights into the performance and robustness of our model.

A comprehensive review process is undertaken to better understand the root causes of misclassification. This encompasses examining both the ground truth labels and predicted labels, as well as closely inspecting the CAMs for all B-scans within the mislabeled cases. Several noteworthy observations emerge from this review, underscoring the exceptional capabilities of our model. Firstly, we identify discrepancies in ground truth labeling, where entire volumes were inaccurately labeled. For instance, in the case of “NORMAL_34”, 22 B-scans were erroneously labeled as drusen, despite the volume being designated as “NORMAL”. Subsequent rectifications are made to resolve these labeling errors throughout the dataset. Secondly, instances were found where our model accurately detected the presence of drusen, while the dataset erroneously labeled these cases as normal. This expertise plays a pivotal role in identifying and rectifying such discrepancies. Lastly, it was acknowledged that some images in the dataset posed significant challenges due to excessive noise or the presence of conditions such as cataracts. To address this, problematic cases were excluded from further analysis. As a result of these corrections, our model’s accuracy rate substantially improves to 97%, with only 15 remaining misclassified cases. These developments significantly enhance the robustness and reliability of our proposed model and underscore the importance of close collaboration between medical experts and machine learning practitioners in the realm of healthcare AI research.

In summary, our paper has introduced an innovative ensemble deep learning model approach for the classification of age-related macular degeneration (AMD) using optical coherence tomography (OCT) images. We conducted a thorough analysis of renowned deep learning models, including ResNet, EfficientNet, and Attention mechanisms, and amalgamated their predictive strengths into a unified ensemble model. Our methodology was rigorously evaluated on the Noor dataset, encompassing 16,822 retinal images from 441 patients across “Normal”, “Drusen”, and “Choroidal Neovascularization (CNV)” classes, aligning with established categories in pertinent literature.

The significance of our ensemble model lies in its pursuit of heightened accuracy and performance in OCT image classification, charting a course for a comprehensive advancement in ophthalmic diagnosis and treatment strategies. Notably, our ensemble model surpassed individual models in precision, recall, F1 score, and overall accuracy across all AMD classes, demonstrating the effectiveness of combining multiple models to leverage their collective intelligence for precise classification. Crucial to our success were data pre-processing and augmentation methods, including resizing, normalization, random cropping, flipping, rotation, transformation, brightness adjustment, and scaling. Leveraging transfer learning with pre-trained models on large-scale image datasets facilitated model convergence, endowing our models with valuable feature representations.

Addressing the challenge of accurate labeling in medical applications, we implemented an automated system for identifying potentially mislabeled samples, rectifying discrepancies in ground truth labeling through collaborative error analysis. This effort significantly improved the overall accuracy of our model. Utilizing class activation M+maps (CAMs) provided insightful visualizations of regions influencing model predictions, aiding in the interpretation of decision-making processes, and affirming the models’ focus on relevant pathological features.

Our study’s implications for ophthalmology and healthcare AI are substantial, as the proposed ensemble model holds promise for enhancing the accuracy of AMD classification from OCT images, potentially revolutionizing early diagnosis and treatment planning. The increased accuracy can positively impact patient outcomes. Interpretability tools such as CAMs can bolster clinicians’ trust in AI systems by providing insights into model decisions, underscoring the importance of transparency and explainability.

## 4. Conclusions

In conclusion, our research contributes a comprehensive exploration of AI-driven advancements in AMD classification using OCT images. Through a meticulous analysis of models, data augmentation, and error rectification, our study underscores the potential of AI as a valuable tool in ophthalmology. The proposed ensemble model, leveraging ResNet, EfficientNet, and Attention mechanisms, showcases superior accuracy, emphasizing the benefits of model fusion for complex medical image analysis. Our findings stress the critical role of high-quality data and expert collaboration in developing robust AI solutions for healthcare. As AI in healthcare progresses, focusing on interpretability, generalization, and ethical considerations, our research contributes to the ongoing journey towards improved diagnostic accuracy and patient care.

## Figures and Tables

**Figure 1 bioengineering-11-00300-f001:**
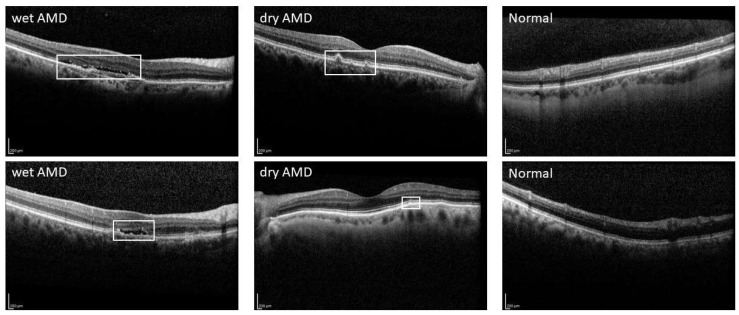
Sample OCT images of wet AMD, dry AMD, and normal eye.

**Figure 2 bioengineering-11-00300-f002:**
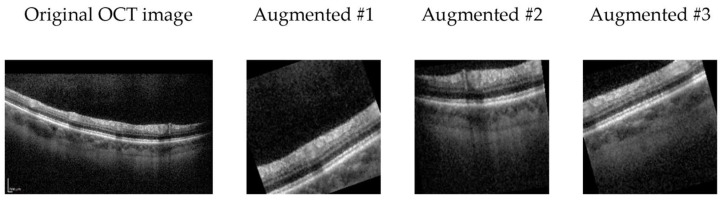
Exemplary OCT B-scan image alongside three augmented variants generated through random transformers.

**Figure 3 bioengineering-11-00300-f003:**
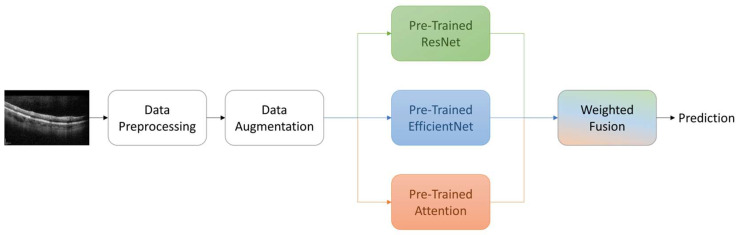
Overview of the proposed methodology.

**Figure 4 bioengineering-11-00300-f004:**
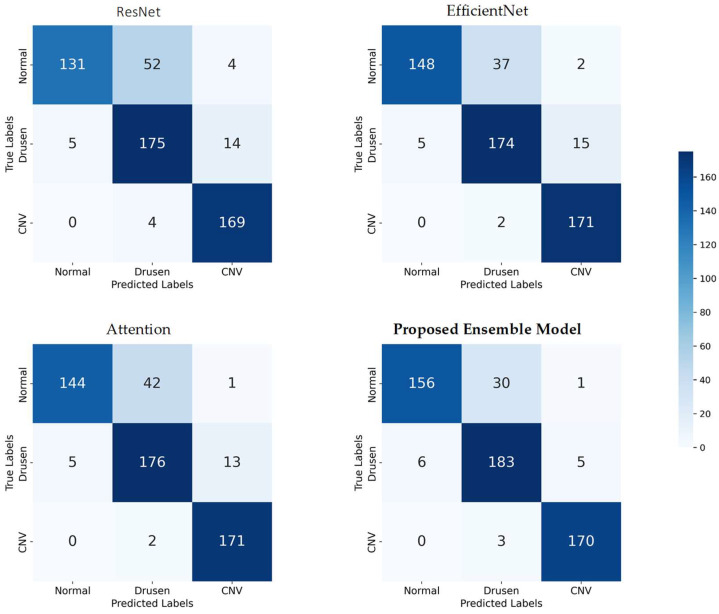
Confusion matrices for model evaluation: true vs. predicted labels across three AMD classes.

**Figure 5 bioengineering-11-00300-f005:**
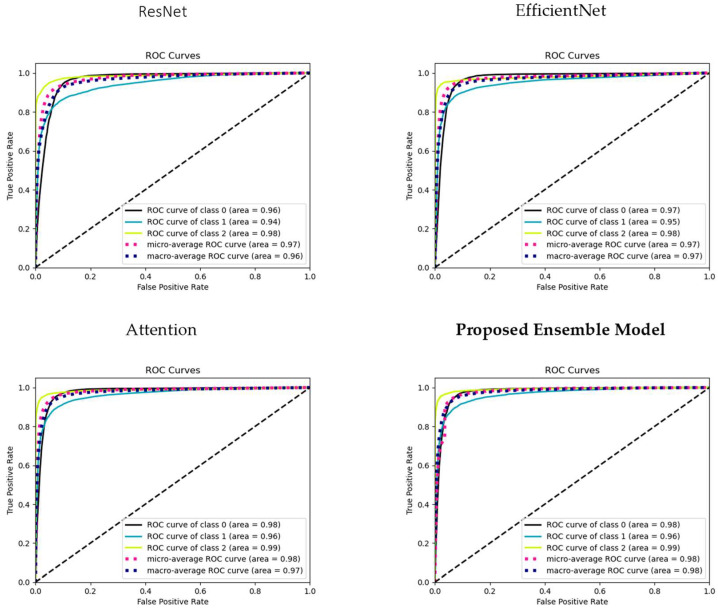
ROC curves illustrating model performance: individual class ROC curves (Classes 0, 1, 2), micro-average ROC curve, and macro-average ROC Curve. The dashed diagonal line represents a reference line (line of no-discrimination).

**Figure 6 bioengineering-11-00300-f006:**
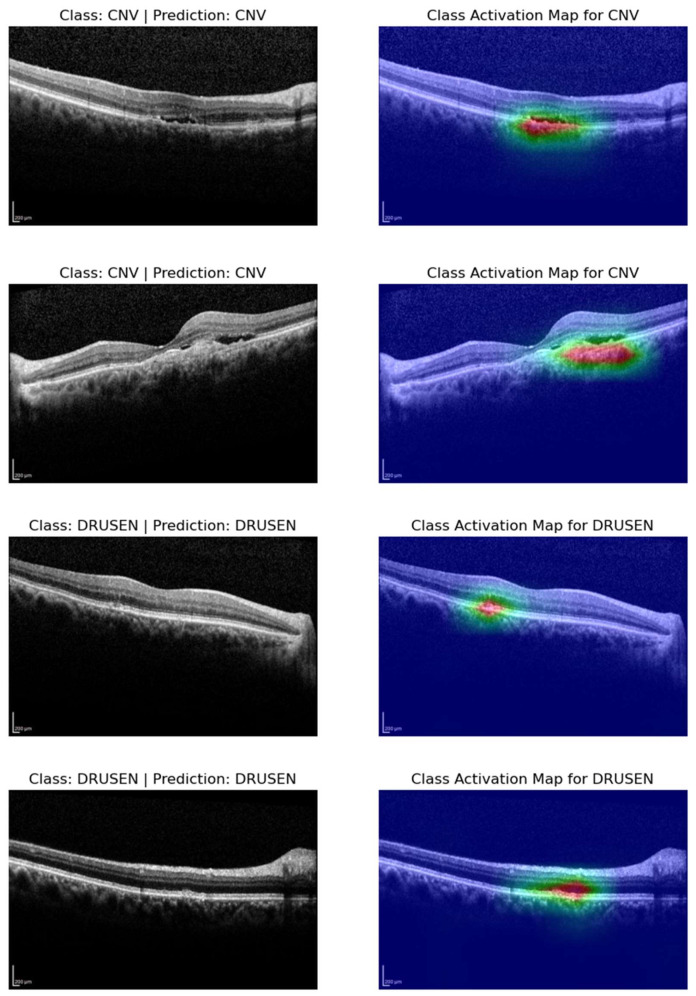
Visualizing class activation maps (CAMs) for representative OCT slices: cases of CNV and drusen classification.

**Table 1 bioengineering-11-00300-t001:** Performance Metrics of Classification Models for Normal, Drusen, and CNV Categories. Best results are highlighted in bold.

Dataset	Method	Class	Precision (%)	Recall (%)	F1 Score (%)	OP (%)	OR (%)	OF (%)	OA (%)
Noor Dataset 5-Fold Cross Validation	ResNet	Normal	96.32	70.05	81.11				
		Drusen	75.76	90.21	82.35	87.26	85.74	86.54	85.54
		CNV	90.37	97.69	93.89				
	EfficientNet	Normal	**96.73**	79.14	87.06				
		Drusen	81.69	89.69	85.50	89.66	88.99	88.91	88.99
		CNV	90.96	**98.84**	94.74				
	Attention	Normal	96.64	77.01	85.71				
		Drusen	80.00	90.72	85.02	89.50	88.63	88.54	88.63
		CNV	92.43	**98.84**	95.53				
	Proposed Ensemble Model	Normal	96.30	**83.42**	**89.40**				
		Drusen	**84.72**	**94.33**	**89.27**	**92.54**	**92.01**	**92.03**	**91.88**
		CNV	**96.59**	98.27	**97.42**				

## Data Availability

The data presented in this study are openly available in: https://github.com/jodeiri/An-Ensemble-Deep-Learning-Model-for-AMD-Classification-using-OCT-images.git (accessed on 10 February 2024).

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
