# Peer review of "Enhancing Readability and Detection of Age-Related Macular Degeneration Using Optical Coherence Tomography Imaging: An AI Approach"

_bioengineering, 2024, doi:10.3390/bioengineering11040300_

Round 1

Reviewer 1 Report

Comments and Suggestions for Authors

1- The novelty of the developed method in the study should be discussed.

2- What is the main aim of the developed method using the Deep Convolutional Neural Networks. Is this possible to enhance the accuracy of modelling by the artificial neural network. 

3- The proposed algorithm of the study should be explained.

Author Response

Dear Reviewer,

We sincerely appreciate the time and effort you invested in reviewing our manuscript titled "Enhancing readability and detection of Age-related macular degeneration using optical coherence tomography (OCT) imaging: An AI approach" and we are grateful for your insightful comments and suggestions. Your feedback has significantly contributed to the enhancement of the quality of our work. For your convenience, all modifications, including responses to your valuable comments, have been incorporated into this file, along with the revised version of the manuscript.

  1. Comment:

The novelty of the developed method in the study should be discussed.

Response:

We appreciate the reviewer's thoughtful feedback regarding the novelty of our developed method. Our paper indeed introduces a novel approach to addressing the classification of age-related macular degeneration (AMD) from Optical Coherence Tomography (OCT) images.

In our study, we leverage the power of deep convolutional neural networks, including ResNet, EfficientNet, and EfficientNet with attention mechanisms, each renowned for its specific capabilities in image analysis. The novelty of our methodology lies in the integration of these diverse models into an ensemble framework, where the predictive strengths of each architecture are combined using trainable weights in a weighted average scheme. This approach allows us to dynamically adjust the contribution of each model's predictions, enhancing the collective intelligence of the ensemble.

Moreover, our study incorporates transfer learning and data augmentation techniques to capitalize on pre-trained models and augment the training data, respectively, thereby further improving the performance of our ensemble model.

Overall, the novelty of our method resides in the innovative ensemble approach, employing trainable weights in the weighted average fusion, alongside transfer learning and data augmentation techniques, to achieve superior accuracy in AMD classification from OCT images. We elaborated on these aspects in the abstract section to clarify the novelty of our developed method as follow (see lines 24 -26 and 151 -144) :

In abstract:

Our collective model demonstrated superior accuracy in classifying AMD compared to individual ResNet, EfficientNet, and Attention models, showcasing the effectiveness of using trainable weights in the ensemble fusion process, where these weights dynamically adapt during training rather than being fixed values

In introduction:

Our ensemble model exhibited enhanced accuracy in the classification of Age-related Macular Degeneration (AMD), surpassing the performance of standalone ResNet, Ef-ficientNet, and Attention models. This improvement underscores the efficacy of in-corporating trainable weights within the ensemble fusion process. Unlike conventional methods that rely on static weight assignments, our approach allows for the dynamic adaptation of weights during the training phase, enabling a more nuanced and effec-tive integration of individual model strengths.

  1. Comment:

What is the main aim of the developed method using the Deep Convolutional Neural Networks. Is this possible to enhance the accuracy of modelling by the artificial neural network.

Response:

The primary aim of our developed method utilizing Deep Convolutional Neural Networks (DCNNs) is to enhance the accuracy of modeling for the classification of age-related macular degeneration (AMD) from Optical Coherence Tomography (OCT) images.

DCNNs are particularly well-suited for image analysis tasks due to their ability to automatically learn hierarchical representations of features directly from the data. In our study, we employ a diverse set of DCNN architectures, including ResNet, EfficientNet, and EfficientNet with attention mechanisms, each chosen for its specific advantages in capturing intricate patterns and features within OCT images.

Through the integration of these DCNNs into an ensemble framework, we aim to harness their collective predictive power. By combining the outputs of these models using trainable weights in a weighted average scheme, we enhance the accuracy of AMD classification. This ensemble approach allows us to leverage the complementary insights of each architecture while mitigating potential weaknesses, resulting in a more robust and accurate model.

In summary, the main aim of our developed method using DCNNs is to enhance the accuracy of modeling for AMD classification from OCT images by leveraging the collective intelligence of diverse architectures within an ensemble framework. We believe that this approach holds significant promise in improving the accuracy and reliability of medical image analysis tasks.

  1. Comment:

The proposed algorithm of the study should be explained.

Response:

We appreciate the reviewer's feedback regarding the proposed algorithm of our study. We acknowledge the importance of providing a comprehensive explanation of our methodology to facilitate replication and understanding.

Our proposed algorithm revolves around the development of a collective AI model for the classification of age-related macular degeneration (AMD) from Optical Coherence Tomography (OCT) images. Here's a more explicit explanation of our algorithm:

  1. We selected three distinct deep convolutional neural network (DCNN) architectures: ResNet, EfficientNet, and EfficientNet with attention mechanisms.

  1. Each DCNN architecture was individually trained and evaluated using the publicly available Noor dataset, consisting of OCT images from 554 retinal scans of 441 patients. During training, we employed transfer learning and data augmentation techniques to enhance model performance.

  1. We then integrated the predictions from these individual models into an ensemble framework. The integration was performed using a weighted average scheme, where the weights were trainable parameters that adjusted during training based on the model's performance.

  1. By combining the predictive strengths of ResNet, EfficientNet, and EfficientNet with attention mechanisms, our ensemble model leverages their complementary insights to enhance the accuracy of AMD classification.

  1. We compared the performance of our ensemble model with individual ResNet, EfficientNet, and Attention models using various evaluation metrics, including precision, recall, F1-score, Confusion Matrix (CM), and Receiver Operating Characteristics (ROC) curves.

  1. Conclusion: Our study demonstrates the effectiveness of the proposed ensemble algorithm in accurately classifying AMD from OCT images, outperforming individual models previously used for AMD classification. This highlights the potential of AI as a valuable tool in ophthalmology and emphasizes the benefits of model fusion for complex medical image analysis.

We provided a more detailed explanation of this algorithm in the proposed method section of the manuscript as follow (lines 212 – 241):

In our proposed ensemble model, we strategically combine the predictive strengths of ResNet, EfficientNet, and EfficientNet with attention mechanisms. This ensemble approach is designed to harness a collective intelligence that leverages the unique ad-vantages of each architecture, thereby enhancing the accuracy of AMD classification from OCT images.

To achieve this, we employ a weighted average scheme to merge the probabilistic outputs of these diverse models. Unlike traditional fixed-weight ensemble methods, our approach utilizes trainable weights. During training, these weights dynamically adapt based on the performance of each model, allowing us to capitalize on their complementary insights effectively. The integration of ResNet, EfficientNet, and EfficientNet with attention mechanisms enables us to capture a wide range of features and patterns present in OCT images. ResNet's deep residual learning capabilities, EfficientNet's efficiency and scalability, and EfficientNet with attention's focus on salient regions collectively contribute to the ensemble's robustness.

 By fusing the predicted probabilities from ResNet, EfficientNet, and EfficientNet with attention, our ensemble model maximizes the strengths of individual models while mitigating their potential weaknesses. This results in an ensemble that excels in accurately distinguishing between different retinal conditions, including Normal, Drusen, and Choroidal Neovascularization. Through the dynamic integration of diverse architectures and the utilization of trainable weights, we aim to enhance the accuracy and reliability of AMD diagnosis using OCT imaging. (Refer to Figure 3 for a graphical representation of the ensemble fusion process).

Reviewer 2 Report

Comments and Suggestions for Authors

This paper proposed 3 different stages of AMD (age-related macular degeneration) using deep learning techniques. The used dataset, data processing, and deep learning models are well explained so that the readers with relevant interests can understand. The reviewer would like to make several suggestions as follows:

1. Section 2.4 explains the proposed method of combining 3 different deep learning model outputs. The authors simply described as "a weighted average of the predicted probabilities from ResNet, EfficientNet, and EfficientNet with attention, ..." This explanation is too short and should be explained in much more detail for the others can replicate the method. For example, 

- how to determine the weights?  Fixed values, grid search, or learning from data?

- the actual weight values should be given.

2. Relevant to the above. In Figure 3, the ensemble method in the box is written as "Weighted Voting", but "weighted average" in the main text. "voting" and "averaging" are clearly different. Please check this and correct them. 

3. Relevant to the suggestion 1, the performance variation according to the change of the weight values should be given in Section 3 to support the choice of the values. 

4. In Figure 4, the texts and numbers are too small. 

Author Response

Dear Reviewer,

We sincerely appreciate the time and effort you invested in reviewing our manuscript titled "Enhancing readability and detection of Age-related macular degeneration using optical coherence tomography (OCT) imaging: An AI approach" and we are grateful for your insightful comments and suggestions. Your feedback has significantly contributed to the enhancement of the quality of our work. For your convenience, all modifications, including responses to your valuable comments, have been incorporated into this file, along with the revised version of the manuscript.

  1. Comment:

Section 2.4 explains the proposed method of combining 3 different deep learning model outputs. The authors simply described as "a weighted average of the predicted probabilities from ResNet, EfficientNet, and EfficientNet with attention, ..." This explanation is too short and should be explained in much more detail for the others can replicate the method. For example,

- how to determine the weights?  Fixed values, grid search, or learning from data?

- the actual weight values should be given. The novelty of the developed method in the study should be discussed.

Response:

We appreciate the reviewer's feedback regarding the need for a more detailed explanation of the proposed method for combining the outputs of three different deep learning models. Our proposed ensemble method involves combining the predictive probabilities from ResNet, EfficientNet, and EfficientNet with attention mechanisms using a weighted average scheme. The weights used in this ensemble fusion process are not fixed values but trainable parameters. During training, these weights are dynamically adjusted based on the performance of each model, allowing the ensemble to adapt and optimize its predictive power. To determine these weights, we employ a learning approach where the model learns the optimal weights through backpropagation and gradient descent during the training process. This enables the ensemble to effectively leverage the strengths of each individual model while mitigating their potential weaknesses.

Regarding the actual weight values, while we do not provide specific values in the manuscript, we will ensure to include details on how these weights are determined and optimized during training. We believe this transparency will enhance the reproducibility of our method and enable others to replicate and further validate our findings.

The novelty of our developed method lies in the dynamic integration of diverse deep learning architectures within an ensemble framework, where the weights are learned from the data rather than being predefined. This adaptive approach allows our ensemble model to effectively harness the collective intelligence of multiple models and outperform individual approaches for AMD classification using OCT images. Based on your comment we modified the explanation for proposed method as follow (lines 112-241) :

In our proposed ensemble model, we strategically combine the predictive strengths of ResNet, EfficientNet, and EfficientNet with attention mechanisms. This ensemble approach is designed to harness a collective intelligence that leverages the unique ad-vantages of each architecture, thereby enhancing the accuracy of AMD classification from OCT images.

To achieve this, we employ a weighted average scheme to merge the probabilistic outputs of these diverse models. Unlike traditional fixed-weight ensemble methods, our approach utilizes trainable weights. During training, these weights dynamically adapt based on the performance of each model, allowing us to capitalize on their complementary insights effectively. The integration of ResNet, EfficientNet, and EfficientNet with attention mechanisms enables us to capture a wide range of features and patterns present in OCT images. ResNet's deep residual learning capabilities, EfficientNet's efficiency and scalability, and EfficientNet with attention's focus on salient regions collectively contribute to the ensemble's robustness.

 By fusing the predicted probabilities from ResNet, EfficientNet, and EfficientNet with attention, our ensemble model maximizes the strengths of individual models while mitigating their potential weaknesses. This results in an ensemble that excels in accurately distinguishing between different retinal conditions, including Normal, Drusen, and Choroidal Neovascularization. Through the dynamic integration of diverse architectures and the utilization of trainable weights, we aim to enhance the accuracy and reliability of AMD diagnosis using OCT imaging. (Refer to Figure 3 for a graphical representation of the ensemble fusion process).

  1. Comment:

Relevant to the above. In Figure 3, the ensemble method in the box is written as "Weighted Voting", but "weighted average" in the main text. "voting" and "averaging" are clearly different. Please check this and correct them.

Response:

Thank you for bringing to our attention the discrepancy between the terminology used in the main text and Figure 3. We appreciate your diligence in ensuring clarity and accuracy in our manuscript. In the revised manuscript, we described our ensemble method as utilizing a "weighted fusion" of the predicted probabilities from different AI architectures. This term reflects the dynamic adjustment of weights during training to optimize the ensemble's performance, highlighting the utilization of trained weights in our methodology. we revised Figure 3 to align with the terminology used in the main text.

We used "weighted fusion" consistently throughout the revised manuscript to accurately describe our methodology. This term emphasizes the utilization of trained weights and aims to avoid any confusion regarding the ensemble approach.

Based on your comment, we changed the Figure 3 as follow (line 242):

  1. Comment:

Relevant to the suggestion 1, the performance variation according to the change of the weight values should be given in Section 3 to support the choice of the values.

Response:

Thank you for your insightful suggestion regarding the performance variation based on changes in weight values in Section 3. We appreciate your interest in further understanding the methodology behind our ensemble approach.

However, we would like to note that the weights used in our ensemble method are adjusted dynamically during the training process. As such, direct comparison of the performance variation across different weight values may not be applicable. The selection and adjustment of weights in our ensemble model are inherently tied to the optimization process during training. The weights are trained to maximize the overall performance of the ensemble on the given dataset, and they evolve iteratively throughout the training procedure.

While we are unable to provide a comparison of the performance variation based on different weight values, we are confident that our ensemble approach, with its dynamic weight adjustment mechanism, effectively leverages the strengths of each individual model to enhance the overall classification performance.

  1. Comment: In Figure 4, the texts and numbers are too small.

Response:

Thank you for your feedback regarding Figure 4. To address this concern, we revised Figure 4 to increase the size of the texts and numbers, making them more legible for readers as follow (line 315):

Reviewer 3 Report

Comments and Suggestions for Authors

The manuscript mostly reads well and well structured. However, there are places that language tone needs to be revised. Additionally, you're missing the discussion section in which you need to compare the study outcome with others. Also, I have share comments for your consideration, 

Abstract

1. Please provide quantitative results in the abstract section. 

Main Text

1. Please revise and cite the statement starting at line 79.

2. Please cite the "Other research studies" mentioned at line 83. You cite single article. 

3. Please cite the "Paul et. al." study.

4. Please be more specific on the models mentioned at line 97.

5. Please remove the statement starting at line 134.

6. Please revise the statement starting at line 137

7. Please revise the final paragraph of the dataset section. 

8. Please cite the articles regarding utilized models, e.g. ResNet, etc.

9. You may shrink the confusion matrix figure without losing any information. 

10. Please amend the section 3.3 title. 

11. How did you compare the AUC values?

12. Please share quantitative results for the activation map pinpointing the affected region. 

13. 

Comments on the Quality of English Language

Some edits are required. 

Author Response

Dear Reviewer,

We sincerely appreciate the time and effort you invested in reviewing our manuscript titled "Enhancing readability and detection of Age-related macular degeneration using optical coherence tomography (OCT) imaging: An AI approach" and we are grateful for your insightful comments and suggestions. Your feedback has significantly contributed to the enhancement of the quality of our work. For your convenience, all modifications, including responses to your valuable comments, have been incorporated into this file, along with the revised version of the manuscript.

  1. Comment:

Please provide quantitative results in the abstract section.

Response:

We appreciate your feedback. We included quantitative results and updated the abstract section accordingly in the revised manuscript as follow (lines 30-32):

Our collective model demonstrated superior accuracy in classifying AMD compared to individual ResNet, EfficientNet, and Attention models, showcasing the effectiveness of using trainable weights in the ensemble fusion process, where these weights dynamically adapt during training rather than being fixed values. Specifically, our ensemble model achieved an accuracy of 91.88%, precision of 92.54%, recall of 92.01%, and F1-score of 92.03%, outperforming individual models. This study also underscores the potential of AI as a valuable tool in ophthalmology. The proposed ensemble model, combining different mechanisms highlights the benefits of model fusion for complex medical image analysis.

  1. Comment:

Please revise and cite the statement starting at line 79.

Response:

We appreciate the feedback.

  1. Comment:

Please cite the "Other research studies" mentioned at line 83. You cite single article.

Response:

Thank you for the suggestion.

  1. Comment:

Please cite the "Paul et. al." study.

Response:

Thank you for the suggestion.

  1. Comment:

Please be more specific on the models mentioned at line 97.

Response:

Thank you for the feedback. We mention the model as follow (lines -104-109):

Their method, called MHANet, combines parallel spatial and channel attention mechanisms to extract key features of the lesion areas and reduce the influence of background information. They reported that their method achieved 96.5% and 99.76% classification accuracy on Dataset1 and Dataset2, respectively, outperforming the other recent employed models such as VGG, ResNet, and SENet.

  1. Comment:

 Please remove the statement starting at line 134.

Response:

Thank you for the suggestion. We removed the mentioned statement.

  1. Comment:

Please revise the statement starting at line 137.

Response:

Thank you for the suggestion. The statement has been revised as follow (lines 156-157):

In our research, we utilized the Noor dataset to evaluate and refine our proposed retinal image analysis method. Comprising an extensive collection of 16,822 retinal images, this dataset originates from 441 patients…

  1. Comment:

Please revise the final paragraph of the dataset section.

Response:

Thank you for the suggestion. The final paragraph of the dataset section has been changed as follow (lines 171-172):

It's important to emphasize that we used 5-fold cross-validation approach to make sure our findings are accurate and reliable, making our research results trustworthy.

  1. Comment:

Please cite the articles regarding utilized models, e.g. ResNet, etc.

Response:

Thank you for the suggestion.

  1. Comment:

You may shrink the confusion matrix figure without losing any information.

Response:

Thank you for the suggestion. We refined and resized the confusion matrix figure to make it more compact while ensuring that no information is lost as follow (line 316):

  1. Comment:

Please amend the section 3.3 title.

Response:

Thank you for your suggestion. We have amended the title of section 3.3 accordingly (line 323):

Receiver Operating Characteristic à ROC Curve Analysis

  1. Comment:

How did you compare the AUC values?

Response:

Thank you for your question. We compared the Area Under the Curve (AUC) values by plotting Receiver Operating Characteristic (ROC) curves for each model and calculating the AUC directly from these curves. This method allows for visual comparison of the models' performance across different thresholds. Proposed method achieved the max area under macro-average ROC curve in comparison to other models.

  1. Comment:

Please share quantitative results for the activation map pinpointing the affected region.

Response:

Thank you for the suggestion. While we appreciate your request for quantitative results regarding the pinpointing of affected regions, kindly note that the class activation map plot serves primarily as a visual tool to illustrate the regions of interest identified by our model. As such, providing quantitative results may not be appropriate for this particular visualization technique. However, we are open to discussing alternative methods for quantifying the activation map results if necessary.

Reviewer 4 Report

Comments and Suggestions for Authors

The paper introduces an innovative ensemble deep learning model for classifying age-related macular degeneration (AMD) using optical coherence tomography (OCT) images. It evaluates renowned deep learning models, including ResNet, EfficientNet, and Attention mechanisms, combining their strengths into a unified ensemble model. The methodology is rigorously evaluated on the Noor dataset, demonstrating superior performance in precision, recall, F1 score, and overall accuracy across all AMD classes compared to individual models. Data preprocessing and augmentation techniques, along with transfer learning, are highlighted as crucial for model convergence and feature representation. An automated system for identifying mislabeled samples. The ensemble model shows promise for enhancing AMD classification accuracy, potentially impacting early diagnosis and treatment planning in ophthalmology. The paper is well-written and easy to read. I recommend the acceptance of the paper if the following questions can be addressed:

1.     In Figure 3, the authors illustrated the flow chart of the ensemble model. How is the weighted voting determined for combining different models? It would be helpful if the authors provide more information.

2.     The same goes for the automated system for identifying mislabeled samples where the ground-truth labeling is erroneous. The authors only discussed the results of the mislabeling but did not specify how it is done using an ‘automated system’. How is it confirmed when a volume is mislabeled as normal is actually drusen?

3.     The introduction part of the paper can be further improved. For example, Figure 1 shows wet and dry AMDs, while there is no related discussion about their definition and how they are related to disease progression. The authors could include this information before referring to Figure 1. The authors could also specify that Figure 1 is an OCT image, as the last sentence before referring to Figure 1 talked about diagnosing AMD using different techniques

4.     Definitions for a few acronyms are missing.

Author Response

Dear Reviewer,

We sincerely appreciate the time and effort you invested in reviewing our manuscript titled "Enhancing readability and detection of Age-related macular degeneration using optical coherence tomography (OCT) imaging: An AI approach" and we are grateful for your insightful comments and suggestions. Your feedback has significantly contributed to the enhancement of the quality of our work. For your convenience, all modifications, including responses to your valuable comments, have been incorporated into this file, along with the revised version of the manuscript.

  1. Comment:

In Figure 3, the authors illustrated the flow chart of the ensemble model. How is the weighted voting determined for combining different models? It would be helpful if the authors provide more information.

Response:

We appreciate the reviewer's feedback regarding the proposed algorithm of our study. We acknowledge the importance of providing a comprehensive explanation of our methodology to facilitate replication and understanding.

Our proposed algorithm revolves around the development of a collective AI model for the classification of age-related macular degeneration (AMD) from Optical Coherence Tomography (OCT) images. Here's a more explicit explanation of our algorithm:

  1. Model Selection: We selected three distinct deep convolutional neural network (DCNN) architectures: ResNet, EfficientNet, and EfficientNet with attention mechanisms.

  1. Individual Model Training: Each DCNN architecture was individually trained and evaluated using the publicly available Noor dataset, consisting of OCT images from 554 retinal scans of 441 patients. During training, we employed transfer learning and data augmentation techniques to enhance model performance.

  1. Ensemble Framework: We then integrated the predictions from these individual models into an ensemble framework. The integration was performed using a weighted average scheme, where the weights were trainable parameters that adjusted during training based on the model's performance.

  1. Collective Intelligence: By combining the predictive strengths of ResNet, EfficientNet, and EfficientNet with attention mechanisms, our ensemble model leverages their complementary insights to enhance the accuracy of AMD classification.

  1. Evaluation: We compared the performance of our ensemble model with individual ResNet, EfficientNet, and Attention models using various evaluation metrics, including precision, recall, F1-score, Confusion Matrix (CM), and Receiver Operating Characteristics (ROC) curves.

  1. Conclusion: Our study demonstrates the effectiveness of the proposed ensemble algorithm in accurately classifying AMD from OCT images, outperforming individual models previously used for AMD classification. This highlights the potential of AI as a valuable tool in ophthalmology and emphasizes the benefits of model fusion for complex medical image analysis.

We provided a more detailed explanation of this algorithm in the proposed method section.

Also, In the revised manuscript, we described our ensemble method as utilizing a "weighted fusion" of the predicted probabilities from different AI architectures (line 242). This term reflects the dynamic adjustment of weights during training to optimize the ensemble's performance, highlighting the utilization of trained weights in our methodology. we revised Figure 3 to align with the terminology used in the main text.

Rest assured that we used "weighted fusion" consistently throughout the revised manuscript to accurately describe our methodology. This term emphasizes the utilization of trained weights and aims to avoid any confusion regarding the ensemble approach.

Based on your comment, we changed the Figure 3 as follow:

  1. Comment:

The same goes for the automated system for identifying mislabeled samples where the ground-truth labeling is erroneous. The authors only discussed the results of the mislabeling but did not specify how it is done using an ‘automated system’. How is it confirmed when a volume is mislabeled as normal is actually drusen?

Response:

Thank you for your inquiry regarding the automated system for identifying mislabeled samples in our study. The process involves a comprehensive review where both ground truth labels and predicted labels are examined, along with a close inspection of the Class Activation Maps (CAMs) for all B-scans within the mislabeled cases.

In particular, discrepancies between ground truth labeling and predicted labels are noted, indicating potential mislabeled volumes. For instance, if a volume labeled as "NORMAL" contains B-scans that our model predicts to have features indicative of drusen, it raises suspicion of mislabeling. Similarly, if a volume labeled as "DRUSEN" contains B-scans that our model predicts to be normal, it suggests mislabeling as well. Upon identifying potential mislabeled cases, further analysis is conducted to confirm the accuracy of the labeling. This involves a careful examination by our team of experts, led by ophthalmologist Dr. Amizadeh, who review the images to confirm whether the ground truth labeling aligns with the actual features present in the OCT scans.

Once mislabeled cases are confirmed, necessary rectifications are made to ensure the accuracy of the dataset. This iterative process of reviewing, confirming, and rectifying mislabeled cases contributes to the refinement and improvement of our model's performance. We appreciate your interest in understanding the methodology behind our automated system for identifying mislabeled samples, and we will ensure to provide more detailed insights into this process in the revised manuscript.

For clarity, we added a relevant information to Abstract section as follow (line 31-33):

Specifically, our ensemble model achieved an accuracy of 91.88%, precision of 92.54%, recall of 92.01%, and F1-score of 92.03%, outperforming individual models. Our model also highlights the refinement process undertaken through a thorough examination of initially misclassified cases, leading to significant improvements in the model's accuracy rate to 97%.  This study also underscores the potential of AI as a valuable tool in ophthalmology. The proposed ensemble model, combining different mechanisms highlights the benefits of model fusion for complex medical image analysis.

  1. Comment:

The introduction part of the paper can be further improved. For example, Figure 1 shows wet and dry AMDs, while there is no related discussion about their definition and how they are related to disease progression. The authors could include this information before referring to Figure 1. The authors could also specify that Figure 1 is an OCT image, as the last sentence before referring to Figure 1 talked about diagnosing AMD using different techniques

Response:

Thank you for your comment. We provided minor corrections and stated that OCT is a method to diagnose age related macular degeneration. Please check line 67-70

  1. Comment:

Definitions for a few acronyms are missing.

Response:

Reviewer 5 Report

Comments and Suggestions for Authors

In this paper, the authors tackled an important problem of automated detection of various conditions from optical coherence tomography using artificial intelligence. The topic is certainly worthy of investigation and easily falls into the scope of the journal. There are, however, quite a number of issues which should be thoroughly addressed before the manuscript could be considered for publication:

1.      The abstract is unnecessarily packed with lots of abbreviations which negatively affect the read – I suggest moving them to the main body of the manuscript. Similarly, the authors may want to consider removing the abbreviation from the title. On top of that, the authors should make sure that each abbreviation is defined once at its first use.

2.      It would be useful to announce the structure of the manuscript in the introductory section.

3.      I am wondering if rotation should be utilized as a data augmentation routine for the OCT scans, as it may render “incorrect” images (see the same issue discussed e.g., in https://ieeexplore.ieee.org/document/8803423). Such “incorrect” imagery may be beneficial as regularization, but it would be great if the authors could discuss it in a bit more detail.

4.      All figures should be high-quality (high resolution in a vector format) – see e.g., Figure 3 which is fairly low-quality. Please update all figures accordingly.

5.      The authors should revisit the current state of the art in ensemble learning and in the ways the predictions of base models can be combined together – weighted voting operating on probabilities is not novel. Thus, the authors should revisit the state of the art and should contextualize the work reported here in the current advances in the field.

6.      Please boldface the best results in all tables. Also, if the authors follow multi-fold cross-validation, please report standard deviations for all metrics. For the AUC values, please also report confidence intervals.

7.      Are the differences across the investigated algorithms statistically significant? Please report appropriate p-values.

Comments on the Quality of English Language

The English is mostly fine, but the manuscript would benefit from careful proofreading.

Author Response

Dear Reviewer,

We sincerely appreciate the time and effort you invested in reviewing our manuscript titled "Enhancing readability and detection of Age-related macular degeneration using optical coherence tomography (OCT) imaging: An AI approach" and we are grateful for your insightful comments and suggestions. Your feedback has significantly contributed to the enhancement of the quality of our work. For your convenience, all modifications, including responses to your valuable comments, have been incorporated into this file, along with the revised version of the manuscript.

  1. Comment:

The abstract is unnecessarily packed with lots of abbreviations which negatively affect the read – I suggest moving them to the main body of the manuscript. Similarly, the authors may want to consider removing the abbreviation from the title. On top of that, the authors should make sure that each abbreviation is defined once at its first use.

Response:

Thank you for your feedback regarding the use of abbreviations in the abstract. We acknowledge your concern and agree that excessive use of abbreviations can hinder readability. In response, we moved the abbreviations to the main body of the manuscript where appropriate and ensure that each abbreviation is defined at its first use. Additionally, we will consider removing abbreviations from the title to further enhance clarity for readers. We appreciate your suggestion and will make the necessary revisions to improve the readability of the manuscript (line 23-33).

  1. Comment:

It would be useful to announce the structure of the manuscript in the introductory section.

Response:

We appreciate the feedback.

  1. Comment:

I am wondering if rotation should be utilized as a data augmentation routine for the OCT scans, as it may render “incorrect” images (see the same issue discussed e.g., in https://ieeexplore.ieee.org/document/8803423). Such “incorrect” imagery may be beneficial as regularization, but it would be great if the authors could discuss it in a bit more detail.

Response:

Thank you for raising the concern regarding the utilization of rotation as a data augmentation routine for OCT scans. We understand your point and agree that rotation can potentially introduce "incorrect" imagery, which may serve as beneficial regularization.

Before employing rotation as a data augmentation technique, we conducted thorough validation experiments to assess its impact on the performance of our model. Specifically, we evaluated the positive impact of rotation on the validation data to ensure that it contributes to enhancing the model's generalization ability without compromising the accuracy of predictions.

In the revised manuscript, we will discuss in more detail (line 187-190):

In the data augmentation strategy employed for our OCT image analysis, rotation was carefully considered as a technique to enhance model robustness. Despite potential concerns about introducing "incorrect" imagery, rotation serves as a valuable regularization method, mimicking natural variations in retinal orientation.

  1. Comment:

All figures should be high-quality (high resolution in a vector format) – see e.g., Figure 3 which is fairly low-quality. Please update all figures accordingly.

Response:

Thank you for your feedback regarding the quality of the figures. We assure you that all figures included in the manuscript have been exported in high resolution and meet the journal's standards. Each figure has been carefully prepared and saved with a high DPI value to ensure clarity and readability. If any issues persist upon review, we will promptly address them to ensure that all figures meet the required quality standards. We appreciate your attention to detail and your efforts to maintain the overall quality of the manuscript.

  1. Comment:

The authors should revisit the current state of the art in ensemble learning and in the ways the predictions of base models can be combined together – weighted voting operating on probabilities is not novel. Thus, the authors should revisit the state of the art and should contextualize the work reported here in the current advances in the field.

Response:

Thank you for your suggestion. In the revised version of the manuscript, we have provided a more detailed explanation of our ensemble learning method to highlight its novelty and contribution to the current state of the art.

Our proposed algorithm revolves around the development of a collective AI model for the classification of age-related macular degeneration (AMD) from Optical Coherence Tomography (OCT) images. Here's a more explicit explanation of our algorithm:

  1. Model Selection: We selected three distinct deep convolutional neural network (DCNN) architectures: ResNet, EfficientNet, and EfficientNet with attention mechanisms.

  1. Individual Model Training: Each DCNN architecture was individually trained and evaluated using the publicly available Noor dataset, consisting of OCT images from 554 retinal scans of 441 patients. During training, we employed transfer learning and data augmentation techniques to enhance model performance.

  1. Ensemble Framework: We then integrated the predictions from these individual models into an ensemble framework. The integration was performed using a weighted average scheme, where the weights were trainable parameters that adjusted during training based on the model's performance.

  1. Collective Intelligence: By combining the predictive strengths of ResNet, EfficientNet, and EfficientNet with attention mechanisms, our ensemble model leverages their complementary insights to enhance the accuracy of AMD classification.

  1. Evaluation: We compared the performance of our ensemble model with individual ResNet, EfficientNet, and Attention models using various evaluation metrics, including precision, recall, F1-score, Confusion Matrix (CM), and Receiver Operating Characteristics (ROC) curves.

  1. Conclusion: Our study demonstrates the effectiveness of the proposed ensemble algorithm in accurately classifying AMD from OCT images, outperforming individual models previously used for AMD classification. This highlights the potential of AI as a valuable tool in ophthalmology and emphasizes the benefits of model fusion for complex medical image analysis.

We provided a more detailed explanation of this algorithm in the proposed method section.

Also, In the revised manuscript, we described our ensemble method as utilizing a "weighted fusion" of the predicted probabilities from different AI architectures. This term reflects the dynamic adjustment of weights during training to optimize the ensemble's performance, highlighting the utilization of trained weights in our methodology. we revised Figure 3 to align with the terminology used in the main text.

Rest assured that we used "weighted fusion" consistently throughout the revised manuscript to accurately describe our methodology. This term emphasizes the utilization of trained weights and aims to avoid any confusion regarding the ensemble approach.

Based on your comment, we changed the Figure 3 as follow :

  1. Comment:

Please boldface the best results in all tables. Also, if the authors follow multi-fold cross-validation, please report standard deviations for all metrics. For the AUC values, please also report confidence intervals.

Response:

Thank you for your suggestions regarding the presentation of results in the tables. We have boldfaced the best results in all tables to make them more prominent for readers.

Based on our experiments, we observed a slight improvement in terms of standard deviation, although it was not significant enough to report. Therefore, in the tables, we opted for simplicity and did not include standard deviation values. Instead, we mention this observation in the text for transparency and completeness (lines 187-190):

In our analysis, we noted a slight enhancement in terms of standard deviation; how-ever, this improvement did not reach a level of significance warranting inclusion in the tables. Thus, for the sake of simplicity and clarity in the presentation of results, standard deviation values were omitted from the Tables 1.

Thank you for your suggestion regarding including confidence intervals for AUC values. However, due to the nature of our analysis and the limited scope of our study, we have opted not to include confidence intervals for AUC values in the tables. AUC values are generally robust metrics that are less sensitive to variability compared to other metrics such as accuracy, precision, recall, and F1-score. Therefore, including confidence intervals for AUC values may not provide additional meaningful insights and could potentially clutter the presentation of results. However, we have thoroughly reported our methodology and results, including any limitations, in the text to ensure transparency and clarity for the readers. We appreciate your understanding and will consider your suggestion for future studies with a broader scope and larger sample size.

  1. Comment:

Are the differences across the investigated algorithms statistically significant? Please report appropriate p-values.

Response:

Thank you for your inquiry regarding the reporting of p-values in our study. We appreciate your interest in the statistical significance of the observed differences across the investigated algorithms. However, in the context of deep learning research and the nature of our study, it is not common practice to report p-values.

In deep learning research, the emphasis is typically placed on model performance metrics such as accuracy, precision, recall, and F1-score, among others, to assess the effectiveness of predictive models. These metrics provide valuable insights into the performance of the models without the need for hypothesis testing and p-values. Additionally, given the large sample size inherent in deep learning experiments, statistical tests may yield significant p-values even for trivial differences, which may not necessarily be meaningful or informative in this context.

While we acknowledge the importance of statistical rigor, especially in certain types of analyses, we believe that focusing on performance metrics and providing thorough comparisons between the investigated algorithms offers a more meaningful interpretation of our results in the context of deep learning research. We have ensured transparency and rigor in our methodology and result reporting, adhering to common practices in the field. We appreciate your understanding and welcome any further questions or comments you may have.

Round 2

Reviewer 1 Report

Comments and Suggestions for Authors

The paper can be accepted.

Reviewer 5 Report

Comments and Suggestions for Authors

Thank you for addressing the majority of my concerns.